# A hybrid framework of statistical, machine learning, and explainable AI methods for school dropout prediction

Mst. Rokeya Khatun[1]*, Mithila Akter Mim[1], Md. Mahadi Tasin[2], Md. Minoar Hossain[3]

1 Department of Computer Science and Engineering, Bangladesh University, Dhaka, Bangladesh,
2 RAJUK Uttara Model College, Uttara, Dhaka, Bangladesh, 3 Mawlana Bhashani Science and
Technology University, Tangail, Bangladesh

* rokeya2kcse@gmail.com

## Abstract

Student dropout is a significant challenge in Bangladesh, with serious impacts on both educational and socio-economic outcomes. This study investigates the factors influencing school dropout among students aged 6–24 years, employing data from the 2019 Multiple Indicator Cluster Survey (MICS). The research integrates statistical analysis with machine learning (ML) techniques and explainable AI (XAI) to identify key predictors and enhance model interpretability. Initially, descriptive and inferential statistical analyses were applied to identify significant predictors and guide feature selection. The hybrid feature selection strategy, combining statistical significance and model-based importance measures, revealed key features. Logistic regression was applied to identify statistically significant predictors of school dropout. ML algorithms, Random Forest (RF) and Extreme Gradient Boosting (XGB), were used to build predictive models. Model performance was evaluated using accuracy, precision, recall, and an F1 score. The XGB model achieved the best performance with an accuracy of 94.4%, followed by the RF model. To interpret model predictions and ensure transparency, SHAP (Shapley Additive Explanations) and LIME (Local Interpretable Model-Agnostic Explanations) were employed in tandem with the statistical analyses. Key factors influencing student dropout included age, sex, completed grade, last education grade, division, wealth index, father's and mother's education. These insights offer a data-driven foundation for policymakers to develop targeted intervention strategies aimed at reducing student dropout rates and improving educational outcomes in Bangladesh.

## Introduction

School dropout remains a critical issue affecting educational outcomes globally, particularly among children and young adults [1]. This problem is common in many countries, and dropout rates vary depending on factors like economics, social norms,

**Data availability statement:** The data underlying the results presented in this study are available from the UNICEF Multiple Indicator Cluster Surveys (MICS) database. Access to the dataset requires permission from the UNICEF MICS team due to confidentiality policies. Researchers can request access by contacting the MICS team through their official website: https://mics.unicef.org/surveys.

**Funding:** The author(s) received no specific funding for this work.

**Competing interests:** The authors have declared that no competing interests exist.

and access to essential education resources availability. This is concerning in developing countries, where many students still don't have the same facilities and chances to go to school, leading to higher dropout rates [2]. Reducing school dropout rates is crucial for encouraging comprehensive and equitable education, which is essential for a nation's long-term growth. In the context of Bangladesh, reducing dropout rates is vital for meeting the United Nations Sustainable Development Goals (SDGs), particularly SDG 4, which aspires to provide quality education for all [3]. High dropout rates not only prevent people from completing their education, but they also impede a nation's development by lowering the number of skilled workers, generating significant poverty rates [4]. In several dimensions, it affects families, communities, and the country's general socioeconomic framework [5].

Researchers have conducted various studies using statistical analysis, ML techniques, and Deep learning (DL) techniques to explore the factors influencing student dropout. Key factor identification helps institutions design effective intervention strategies to reduce student dropout. Furthermore, research has shown that data mining techniques can accurately predict student dropout, offering valuable insights into factors like academic performance, socio-economic status, and personal characteristics [6]. Researchers have also investigated the application of ML and DL to develop predictive models for students considered at risk on online learning platforms [7]. Various studies have proved the effectiveness of ML algorithms such as decision tree (DT), naive bayes (NB), K-nearest neighbors (KNN), and linear models in making accurate predictions. For instance, a study using data from 12th graders in South Africa found that the random forest (RF) model achieved the highest accuracy (82%) in predicting student risk profiles [8]. The study [9] shows that the RF algorithm was more accurate than the DT algorithm (78.09% vs. 79.45%), which is a big improvement that indicates how useful advanced preprocessing and feature selection techniques can be. Also, a study [10] looks at how well four ML algorithms, KNN, logistic regression (LR), multi-layer perceptron (MLP), and RF work with Uwezo data from Tanzania to make dropout prediction models better. Furthermore, researchers have significantly improved the predictive model performance by using various ML and DL models to identify dropout rates among university students, particularly after the first semester [11]. Deep learning models have also been applied successfully in dropout prediction, with models such as Long Short-Term Memory and One-Dimensional Convolutional Neural Networks offering superior performance in attrition prediction across educational platforms [12]. Furthermore, a study [13] applied deep learning models such as deep neural networks and accurately predicted student dropout in university settings, achieving strong results with accuracy scores of 72.4% and AUC values of 0.771. Despite these advancements, there remains a significant gap concerning the use of these techniques for all education sectors and nations.

Few prior studies have focused on an in-depth study identifying the critical factors influencing and forecasting school dropout among Bangladeshi children, according to the literature review. This gap highlights the need for research specifically focused on the Bangladeshi context, utilizing nationally representative datasets like the MICS 2019. The MICS 2019 Bangladesh dataset, collected by the Bangladesh Bureau of

Statistics (BBS) and United Nations Children's Fund (UNICEF), offers comprehensive data on child welfare, education, health, and socio-demographic factors. As dropout is a multi-faceted issue affected by socio-economic, demographic, and academic factors, this research seeks to produce actionable insights by identifying the most influential contributors using robust analytical methods. The novelty of this study lies in its hybrid feature selection approach, combining statistical significance tests with ML model-driven insights from SHAP and RF importance scores. This approach ensures the selected features are not only statistically valid but also practically impactful, capturing both linear and nonlinear associations that are often missed in traditional methods. Moreover, it introduces explainable AI into the education domain, making model predictions transparent and interpretable for the general population. The findings of this research will be instrumental in helping educators, policymakers, and stakeholders implement effective strategies aimed at reducing dropout rates and ensuring that more students complete their education. The primary objectives are:

- Using statistical and ML analysis, identify the primary factors influencing school dropout among Bangladeshi students.

- Employ ML models to effectively forecast and predict student dropout.

- Apply XAI techniques, such as SHAP and LIME, to present and enhance the interpretability of the predictive models.

- Generate detailed insights identifying the underlying causes of school dropout to facilitate the enhancement of intervention strategies.

The structure of this paper is organized into five key sections to ensure a comprehensive analysis of school dropout rates in Bangladesh. It presents the dataset, a detailed explanation of the materials and methods applied, and the performance evaluation criteria for the models. It then covers the steps of data preprocessing, followed by the presentation and discussion of results, including policy implications and further scopes of future work. The findings are placed within the context of existing literature, highlighting comparisons with previous studies and addressing the limitations of this work. Finally, the paper concludes the research work.

## Materials and methods

The workflow diagram in Fig 1 depicts the outlines of a comprehensive process for analyzing student dropout using the MICS dataset 2019 Bangladesh. The analysis begins with domain selection, where a relevant dataset is chosen for investigation, with the primary selected domain-based variables. The next step involves data preprocessing, which includes handling

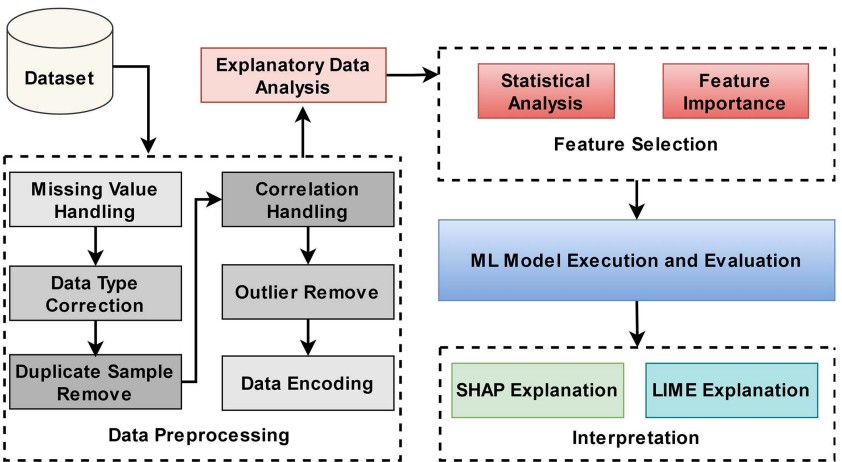

**Fig 1. Workflow diagram for analyzing the key factors that influence the school dropout rate.**

duplicate samples, correcting data types, managing missing values, addressing correlations, and removing outliers to clean the dataset. Once preprocessing is complete, Exploratory Data Analysis (EDA) is performed to reveal patterns and trends that enhance the understanding of the dataset. We then conducted a statistical analysis to explore the relationships between variables and identify feature importance and significance. After that, categorical variables are encoded into numerical values. Furthermore, the ML models are fitted to the dataset, and model evaluation is performed using metrics such as accuracy, precision, recall, and F1 score. Throughout the iterative process, continuous refinement is applied to improve the model's performance by selecting appropriate features and adjusting model parameters. Finally, the best model is identified, and it is further explained using XAI techniques. Whereas SHAP (Shapley Additive Explanations) is utilized to explain the global importance of features, LIME (Local Interpretable Model-Agnostic Explanations) provides local insights into specific predictions.

This study is a secondary analysis of publicly available, fully anonymized data from the UNICEF Multiple Indicator Cluster Surveys (MICS) Bangladesh 2019. The original MICS 2019 data collection was approved by the relevant ethical review boards under the supervision of UNICEF and the Bangladesh Bureau of Statistics (BBS), including the Institutional Review Board (IRB) of BBS. So, no additional ethical approval was required for this secondary analysis. To gain access to the dataset, we needed permission from the UNICEF MICS team, so we reached out to them via email, outlining our research objectives. In response, they provided us with a link to access the dataset. The dataset was accessed on 14 January 2024 for research purposes. The authors had no access to any identifiable participant information during or after data collection.

## Dataset

The MICS 2019 was carried out by BBS and the Bangladesh Ministry of Planning [14], with support from UNICEF. The survey covered a wide range of topics, including health, education, nutrition, water and sanitation, child health, and socio-economic conditions. Data was collected from both urban and rural areas, encompassing all eight divisions and 64 districts. Initially, a sample size of 64,000 households was targeted for the survey. The comprehensive report, "BBS and UNICEF Bangladesh 2019," outlines the survey methodology in detail, including how the sample size was calculated. However, as the aim of this study was to develop predictive models and identify key factors associated with school dropout rather than to derive nationally representative estimates, the survey weights provided by MICS were not applied. While such weights are crucial for calculating population-level statistics, they are not always necessary or appropriate in predictive modeling tasks. From the dataset, we initially selected 11 variables based on expert consultation, prior research, and relevant literature addressing school dropout risk factors in low- and middle-income countries. These variables encompass key determinants from three major domains: demographic characteristics (HL6: age, HL4: sex), socioeconomic status (windex5: wealth index, HH6: area of residence, HH7: administrative division), and familial education background (melevel: mother's education, felevel: father's education, ethnicity: ethnic group). Educational dropout patterns were inferred using ED6 (whether the student completed the grade) and ED16B (the highest grade attained). Additionally, the dependent variable, school attendance (ED9: whether the student attended school in the current year), was used to identify dropout status. According to the dataset distribution, 81.28% of students attended school, while 18.72% did not, indicating a dropout rate of approximately 18.72% among children aged 6–24 years. This figure highlights a significant portion of the school-age population that is not currently enrolled in formal education. Understanding this dropout rate is crucial, as it provides a baseline for model training and helps in identifying the characteristics associated with students at risk of leaving school. In our analysis, we aim to uncover the predictive factors that correlate with this dropout group to inform targeted interventions and policy decisions aimed at improving school retention. Tables 1 and 2 describe the selected variables in detail.

## Statistical measurements

In statistical analysis, understanding the relationship between variables depends upon several key metrics [15]. These metrics, such as odds ratio (OR), p-value, and 95% confidence interval (CI), work together to give a complete picture of how strong, important, and accurate the relationships in the data are [16,17]. The p-value is a statistical measure that

**Table 1. Description of categorical independent variables or features.**

| MICS variables | Variables | Description | Values | Data Type |
|---|---|---|---|---|
| HL4 | Sex | Sex | 'MALE', 'FEMALE' | Nominal |
| HL6 | Age | Student's age | 3-24 years | Numerical |
| ED6 | Completed_grade | Ever completed the grade/year | 'NO', 'YES' | Ordinal |
| ED16B | Last_education_grade | Highest completed grade/year. | 'GRADE 11', 'HSC/Alim/Diploma/Polytechnic', 'GRADE 3', 'GRADE 1', 'GRADE 4', 'GRADE 8', 'GRADE 9', 'GRADE 6', 'GRADE 13', 'GRADE 2', 'SSC/Dakhil', 'GRADE 5', 'GRADE 7', 'GRADE 14' 'BA (Pass)/Fazil', 'BA (Hons)/ MBBS/BSc Engg.' 'MA/MS/MSc and above', 'GRADE 18' | Ordinal |
| HH6 | Area | Type of locality | 'URBAN' 'RURAL' | Nominal |
| HH7 | Division | Geographical/ administrative region | 'Barishal','Chattogram' 'Dhaka','Khulna','Mymenshing','Ra-jshahi','Rangpur', 'Sylhet'. | Nominal |
| windex5 | Wealth_index | Wealth index quintile | 'Richest', 'Fourth', 'Second', 'Poorest', 'Middle', | Ordinal |
| ethnicity | Ethnicity | Ethnic background/ group. | 'Bengali', 'Other' | Nominal |
| melevel | Mother_s_education | Mother's highest education level. | 'Secondary', 'Higher secondary+','Primary', 'Pre-primary or none' | Ordinal |
| felevel | Father_s_education | Father's highest education level. | 'Pre-primary or none', 'Higher secondary+', 'Primary' 'Secondary' | Ordinal |

**Table 2. Description of dependent variable or feature.**

| MICS variables | Variable | Description | Values | Data Type |
|---|---|---|---|---|
| ED9 | Attend_school_thisYear | Attended school during the current school year | 'NO', 'YES' | Ordinal |

indicates whether the relationship between an independent variable and the dependent variable is statistically significant [18]. A low p-value (≤ 0.05) suggests a significant association, while a high p-value (> 0.05) indicates no meaningful link between the variables. An OR greater than 1 indicates increased odds, an OR less than 1 indicates decreased odds, and an OR equal to 1 indicates no effect. For example, an OR of 1.05 means the odds of an outcome increase by about 5% for each unit rise in the predictor, while an OR = 0.70 indicates a 30% reduction in the odds. Similarly, a p-value < 0.05 is treated as evidence of a statistically significant relationship. The 95% CI further aids in understanding this precision. The 95% CI further aids in understanding this precision. If the OR's 95% CI doesn't include 1, it suggests a significant effect of the independent variable on the dependent variable. If the 95% CI for an OR does not include 1, it suggests the result is statistically significant. In this study the binary outcome variable, Attend_school_thisYear, was coded as 1 for children not attending school (indicating dropout) and 0 for those currently attending school. This coding was used consistently across all statistical analyses and machine learning models to ensure interpretability of the results. An OR greater than 1 suggests that the predictor is associated with a higher likelihood of dropout. Conversely, an OR less than 1 indicate a lower likelihood of dropout, meaning the child is more likely to attend school. Odds ratios were interpreted in relation to the reference category or baseline value of each independent variable.

## Machine learning models

Each machine learning (ML) model employs unique methodologies and algorithms. We applied traditional models combined with ensemble techniques, specifically Random Forest (RF) and Extreme Gradient Boosting (XGB). RF is an

ensemble method that aggregates predictions from multiple decision trees, each trained on random data subsets and features. By combining these trees' outputs through averaging or voting, RF reduces overfitting and improves generalization [19,20]. Its ability to handle complex data and multicollinearity while providing interpretable feature importance makes it well-suited for classification tasks. XGB is a powerful gradient boosting framework that builds decision trees sequentially to minimize a differentiable loss function. It improves performance by iteratively correcting errors from previous trees, effectively reducing bias and variance. It also provides internal handling of missing data, regularization to prevent overfitting, and parallel processing for faster training [21,22]. In this study, we applied RF and XGB to analyze an imbalanced dataset where dropouts represent the minority class. To mitigate imbalance effects, class weighting was employed in both models. RF adjusts the weight of classes to balance their influence, while XGB's scale_pos_weight specifically improves sensitivity toward the minority dropout class. These models were chosen due to their proven success in structured data classification and their robustness in handling imbalanced data. Preliminary experiments comparing multiple algorithms confirmed that RF and XGB consistently outperformed others.

Additionally a confusion matrix is a valuable tool for evaluating the performance of a classification model by comparing the model's predicted outcomes with the actual outcomes [23]. It provides a detailed summary of prediction results, which is especially useful in scenarios with imbalanced classes or when different classification errors have varying consequences. By analyzing the matrix, we can gain insights into the types and frequencies of errors the model makes, aiding in understanding its overall performance. Fig 2 depicts the matrix, which includes key terms such as true positive (TP), true negative (TN), false negative (FN), and false positive (FP). Here's a detailed description of an evaluation metric for binary classification problems along with its formulas.

Accuracy: Accuracy measures the proportion of correctly classified instances among all instances in the dataset. It's a commonly used metric for evaluating classification models.

$$\text{Accuracy} = \frac{TP + TN}{TP + TN + FP + FN} \times 100 \qquad (1)$$

**Fig 2. Confusion matrix.**

Positive Predictive Value (PPV) or Precision: It indicates how accurately a model identifies positive cases by measuring the ratio of correctly predicted positives to the total number of positive predictions. It reflects the model's effectiveness in minimizing false positives.

$$\text{Precision} = \frac{TP}{TP + FP} \times 100$$

(2)

Recall (Sensitivity): Recall, or sensitivity, assesses how well a model detects actual positive cases by calculating the ratio of true positives to the total number of real positive instances. It shows the model's capability to identify all relevant positive outcomes.

$$\text{Recall} = \frac{TP}{TP + FN} \times 100$$

(3)

F1_Score: The F1-score represents the harmonic average of precision and recall, offering a balanced evaluation when there's an uneven class distribution. It is particularly useful in situations where both false positives and false negatives are critical.

$$\text{F1\_Score} = 2 \times \frac{Precision \times Recall}{Precision + Recall} \times 100$$

(4)

Negative Predictive Value (NPV): NPV measures the proportion of true negative predictions among all negative predictions made by the model. It quantifies the model's ability to avoid false negatives.

$$\text{NPV} = \frac{TN}{(TN + FN)} \times 100$$

(5)

Receiver Operating Characteristic Area Under Curve (ROC AUC): The ROC AUC evaluates the area beneath the ROC curve, which illustrates the relationship between the true positive rate (sensitivity) and the false positive rate (1-specificity) across various threshold settings [24]. It reflects how effectively a model can distinguish between positive and negative classes at different decision boundaries. The score ranges from 0 to 1, with values nearer to 1 indicating stronger classification performance.

K-fold cross-validation: We have used the K-fold cross-validation method to assess the performance and validity of an ML model [25]. We divide the dataset into k equal parts or folds. The model is then trained and validated k times, using a different fold as the validation set and the remaining $K - 1$ folds as the training set. This process generates K performance scores, which are averaged to produce a single performance estimate. K-fold cross-validation helps reduce overfitting and efficient use of data. This method provides a robust evaluation by leveraging various data subsets for training and validation.

## Explainable AI (XAI)

Explainable AI (XAI) aims to make AI decisions easy to follow [26]. As AI models especially DL ones grow more complex, their inner workings can feel like a 'black box,' hiding how they reach conclusions. XAI provides clear, straightforward explanations of how these models operate and make choices, helping people understand and trust the results.

**SHAP (Shapley Additive Explanations).** SHAP is a methodology based on cooperative game theory that assigns a unique value to each feature, known as the SHAP value [27,28]. This value represents the contribution of that feature to the model's prediction for a specific instance. SHAP is versatile and offers global explanations, which provide insights

into the overall behavior of the model across the entire dataset, and local explanations that clarify how individual features influence specific predictions. SHAP values are consistent, ensuring that the sum of contributions from all features matches the difference between the model's output for a given sample and the average output. While SHAP offers a comprehensive understanding of feature importance and interactions, it can be computationally intensive, particularly for complex models and large datasets.

**LIME (Local Interpretable Model-agnostic Explanations).** LIME explains individual predictions by imitating a complex model's behavior just around that point [29,30]. It does this by creating slight variations of the original example tweaking its feature values and then fitting a simple, easy-to-understand model (like linear regression) on those variations. This lightweight model acts as a local stand-in for the more complicated one near the chosen prediction. Because it works with any machine-learning model and runs quickly, LIME is practical for real-time use. Its downside is that its insights apply only to the small region around that specific prediction and may not hold for the model overall, since it assumes the model behaves in a roughly linear way within that local area.

## Result and discussion

### Data preprocessing

We utilized ML and statistical techniques to streamline our analysis, retaining only the most significant features and narrowing the dataset with key variables. The sample population was defined using age groups from the UNICEF MICS 2019, which categorizes school-going children aged 6–24 years, aligning with global education standards. Children aged 3–5 years were initially included as "school-age" based on the Early Childhood Education (ECE) module of MICS, recognizing this stage as critical for early learning and pre-primary access. Education typically progresses from primary ages (6–10), to lower secondary (11–13), upper secondary (14–15), higher secondary (16–17), and tertiary or vocational training (18–24), though variations may occur due to delayed entry, repetition, or dropout. As ECE or early learning is not formal education and as one cannot drop out of informal education like ECE, school age was defined to be 6–24, excluding the age range of ECE (3–5). Furthermore, to ensure data quality, all samples with missing values in the target variable were excluded. Additionally, records containing placeholder responses such as "Don't know," "No response," or "Missing/DK" in relevant features were removed as well. Initially, the dataset comprised of 48,412 samples; after filtering out these entries, 7,808 rows were discarded, resulting in a final set of 40,604 samples. For categorical features, we replaced missing values with the mode, while for numerical features, we used the mean. After that, we reviewed each feature for mixed data types and converted them into their proper formats casting values to floats, integers, or strings as needed. This step ensured that every column followed a consistent type convention throughout the dataset. Following this, we removed outliers, further refining the dataset to 11 features with 38,213 samples.

To prepare the dataset for ML modeling, we applied a combination of one-hot and label encoding techniques to convert categorical variables into suitable numerical formats. One-hot encoding was used for the nominal categorical features 'Sex', 'Completed_grade', 'Area', 'Division', and 'Ethnicity' with each unique category represented by a separate binary column. To reduce dimensionality and avoid multicollinearity, one category from each feature, used as a reference, was excluded as its presence could be inferred from the remaining columns. This approach prevents the introduction of artificial ordinal relationships into the model. The remaining categorical variables 'Last_education_grade', 'Wealth_index', 'Mother_s_education', and 'Father_s_education' were encoded using label encoding, as they exhibit inherent ordinal relationships. Additionally, 'Age' was retained as a continuous numeric variable representing ages between 6 and 24 years. The binary target variable, 'Attend_school_thisYear', was label encoded as 0 for 'No' and 1 for 'Yes', such that 1 indicates a student attended school this year, while 0 represents a student who did not attend (i.e., a dropout). This systematic encoding ensured that all categorical and numerical variables were appropriately formatted for model training, enabling effective learning while minimizing bias or distortion. Table 3 presents detailed information on the encoding of each feature.

**Table 3. Summary of Encoding Techniques Applied to Dataset Variables.**

| SL | Variables | Encoding Type | Encoded Values |
|---|---|---|---|
| 1 | Sex | One-hot encoding | Separate binary columns for each category: MALE, FEMALE |
| 2 | Completed_grade | One-hot encoding | Binary columns for YES and NO |
| 3 | Last_education_grade | Label encoding | Grade 1 = 1 to Grade 18 = 18 (as per educational level) |
| 4 | Area | One-hot encoding | Binary columns for URBAN and RURAL |
| 5 | Division | One-hot encoding | Binary columns for Barishal, Chattogram, Dhaka, Khulna, Mymensingh, Rajshahi, Rangpur, Sylhet |
| 6 | Wealth_index | Label encoding | Poorest = 0, Second = 1, Middle = 2, Fourth = 3, Richest = 4 |
| 7 | Ethnicity | One-hot encoding | Binary columns for Bengali and Other |
| 8 | Mother_s_education | Label encoding | Pre-primary or none = 0, Primary = 1, Secondary = 2, Higher secondary+ = 3 |
| 9 | Father_s_education | Label encoding | Pre-primary or none = 0, Primary = 1, Secondary = 2, Higher secondary = 3 |
| 10 | Age | | 6-24 years |
| 11 | Attend_school_thisYear | Label encoding | NO = 0, YES = 1 |

Categorical variables were encoded according to their measurement level. For ordinal variables, such as Last_education_grade, Mother_s_education, Father_s_education, and Wealth_index, label encoding was applied to preserve their natural order. These variables reflect a progression in education or socioeconomic status, which is crucial for capturing their influence on educational outcomes. In contrast, nominal variables including Area, Division, Completed_grade, Ethnicity, and Sex do not have an inherent order. Therefore, one-hot encoding was applied to these variables to ensure that each category is treated as independent, preventing any unintended ordinal relationship in the model. This encoding choice aligns with best practices for handling nominal features in machine learning.

**Explanatory data analysis**

Fig 3 presents a violin plot showing the distribution of the numerical feature Age concerning school attendance (Attend_school_thisYear, where Not attending = 0, Attending = 1). The plot reveals that children who are currently attending school tend to be younger, with a concentration between the ages of 6 and 16. In contrast, those not attending school are generally older, with a broader spread from around 10–24, indicating that dropout rates increase with age. The distribution suggests that age is a significant factor influencing school attendance, with older children more likely to leave the education system.

Fig 4 illustrates a set of bar plots that compare the distributions of students who attended school in the reference year versus those who dropped out, across multiple categorical variables labeled numerically according to Table 3. The visualizations reveal several compelling patterns. Firstly, age shows a distinct shift toward higher dropout rates in older age groups, highlighting a tendency for students to leave school as they approach or surpass adolescence. Completed grade and last education grade both show that students with higher educational attainment are significantly less likely to drop out, indicating strong associations between academic progression and retention. Additionally, the division-wise analysis reveals regional disparities, with dropout rates varying notably across administrative divisions such as Sylhet, Barishal, and Chattogram. The wealth index, encoded from 0 (poorest) to 4 (richest), clearly shows that economically disadvantaged students face a higher risk of leaving school, with dropout prevalence decreasing as wealth increases. Both mother's and father's education levels demonstrate similar trends: students whose parents attained secondary or higher education (values 2 and 3, respectively) are more likely to continue their studies, emphasizing the critical role of parental influence in educational outcomes. The ethnicity variable shows that students identified as "Other" (non-Bengali) face a slightly higher dropout risk compared to the majority Bengali group. A mild gender disparity is also visible, with females (2) slightly more prone to dropout than males (1). Lastly, students from rural areas (2) show a marginally higher dropout tendency compared to their urban (1) counterparts, suggesting the influence of geographic and infrastructural challenges.

**Fig 3. Violin plot for numerical feature age.**

Together, these visual insights help inform the feature selection and model-building processes by confirming which socio-demographic attributes have meaningful associations with dropout status. They also underscore the complex, multidimensional nature of school dropout in Bangladesh, justifying the need for hybrid modeling techniques and explainable AI to capture both linear and nonlinear relationships.

### Feature selection

The variance inflation factor (VIF) values were calculated to assess multicollinearity among the predictors. As presented in Table 4, all VIF scores were below the common threshold of 5, indicating low multicollinearity. Division-related variables such as Division_Chattogram (2.59), Division_Dhaka (2.35), and Division_Khulna (2.06) showed relatively higher VIF values compared to others. In contrast, variables like Sex_MALE (1.01) and Area_URBAN (1.12) had the lowest VIFs, suggesting minimal correlation with other predictors. These results support the stability and reliability of the regression model.

Table 5 presents the results of the logistic regression (LR) model, showing that the variables Sex, Age, Completed Grade, Last Education Grade, Father's Education, Wealth Index, Ethnicity, and the regional variable Division_Dhaka were statistically significant predictors. Each exhibited p-values below 0.05 and odds ratios with confidence intervals that do not include 1, indicating strong associations with the outcome variable. As a result, these variables were retained for initial modeling.

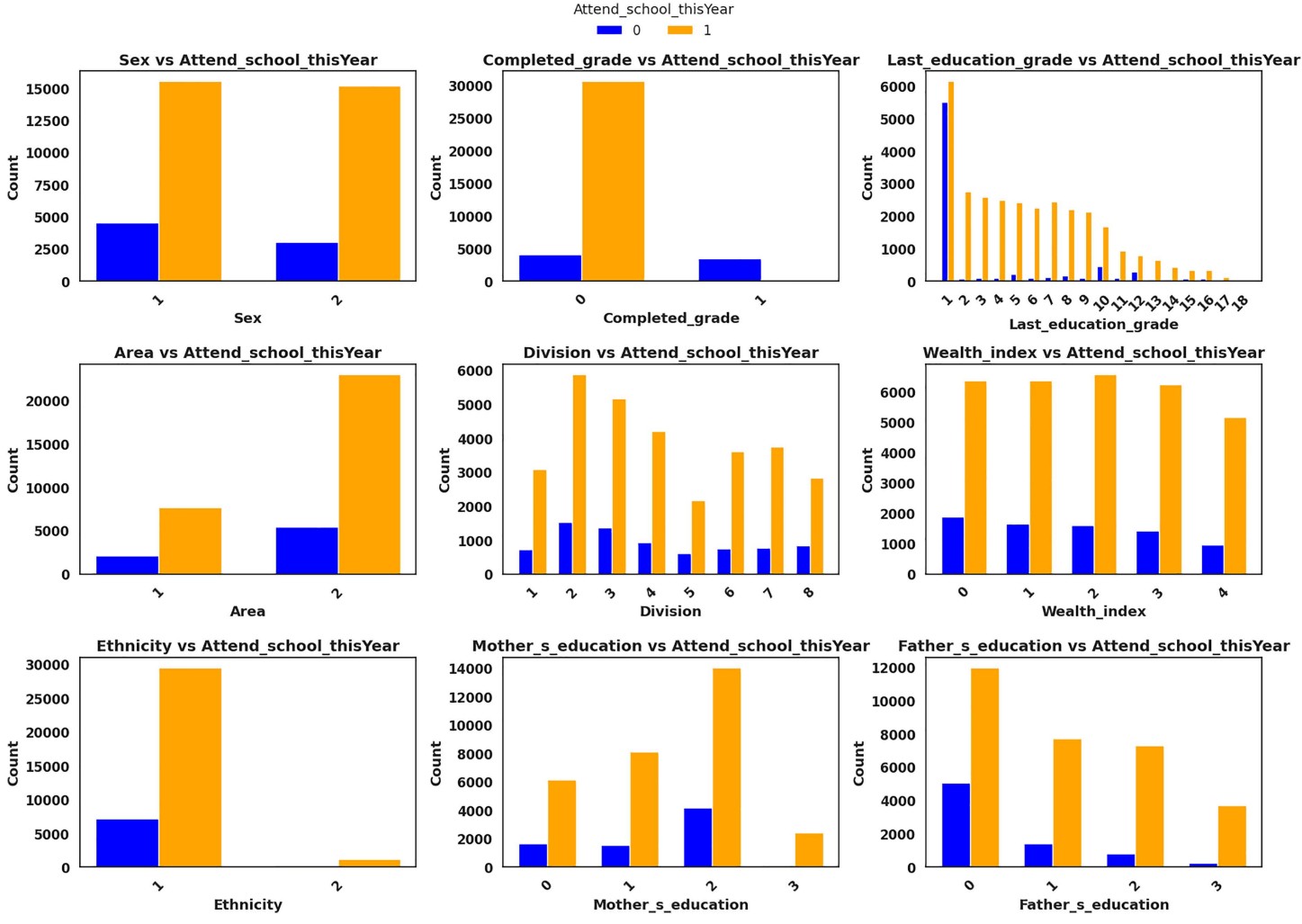

**Fig 4. Countplot for categorical features.**

Notably, Completed Grade (with a p-value of 0.00000 and an odds ratio of 0.00347), Age (p-value 0.00000, odds ratio 0.68252), and Last Education Grade (p-value 0.00000, odds ratio 1.62824) emerged as the most influential predictors. Father's Education and Sex also showed high statistical significance. Additionally, Wealth Index and Ethnicity contributed meaningfully to the model. Among the divisions, only Division_Dhaka was statistically significant, associated with lower odds of the outcome. In contrast, Mother's Education and Area_URBAN were not statistically significant, with p-values of 0.53496 and 0.38166, respectively. Although Ethnicity was statistically significant (p = 0.012) in the LR analysis, it was dropped because it showed very low predictive importance in both the RF and SHAP analyses ranking. This indicates that, despite statistical significance, its practical contribution to model performance was minimal. In contrast, Mother's education, though statistically insignificant (p = 0.536), was retained due to its comparatively higher importance in both ML models and its strong theoretical and contextual relevance. Prior research consistently highlights maternal education as a key determinant of school dropout. Including it helps preserve model interpretability and aligns with domain knowledge, ensuring the model remains meaningful for education-related policy and intervention design. Conversely, Area_URBAN also showed low importance across all methods and was excluded from the final model.

**Table 4. VIF Scores Supporting Low Multicollinearity.**

| Feature | VIF |
|---|---|
| Division_Chattogram | 2.588303 |
| Age | 2.508971 |
| Division_Dhaka | 2.348832 |
| Last_education_grade | 2.087919 |
| Division_Khulna | 2.056806 |
| Division_Rangpur | 1.942607 |
| Division_Rajshahi | 1.920252 |
| Division_Sylhet | 1.836855 |
| Division_Mymenshing | 1.615065 |
| Father_s_education | 1.462482 |
| Wealth_index | 1.354731 |
| Mother_s_education | 1.279299 |
| Completed_grade_YES | 1.213684 |
| Ethnicity_Other | 1.159839 |
| Area_URBAN | 1.116112 |
| Sex_MALE | 1.005760 |

**Table 5. Statistical Analysis of Features Influencing School Dropout.**

| Feature | P-Value | Odds Ratio | CI Lower Bound | CI Upper Bound |
|---|---|---|---|---|
| Age | 0 | 0.68251 | 0.67215 | 0.69304 |
| Last_education_grade | 0 | 1.62824 | 1.598 | 1.65905 |
| Wealth_index | 0.00006 | 1.08568 | 1.04316 | 1.12993 |
| Mother_s_education | 0.53496 | 1.01926 | 0.95964 | 1.08259 |
| Father_s_education | 0 | 1.23176 | 1.16026 | 1.30767 |
| Area_URBAN | 0.38166 | 0.94925 | 0.84472 | 1.06673 |
| Division_Chattogram | 0.15357 | 0.86565 | 0.71005 | 1.05536 |
| Division_Dhaka | 0.00391 | 0.74757 | 0.61352 | 0.91091 |
| Division_Khulna | 0.29922 | 0.8953 | 0.72657 | 1.1032 |
| Division_Mymenshing | 0.5266 | 1.08367 | 0.84505 | 1.38967 |
| Division_Rajshahi | 0.09039 | 1.21237 | 0.97013 | 1.5151 |
| Division_Rangpur | 0.4905 | 1.08012 | 0.86761 | 1.34468 |
| Division_Sylhet | 0.18471 | 0.86094 | 0.69007 | 1.07414 |
| Completed_grade_YES | 0 | 0.00347 | 0.00267 | 0.00452 |
| Ethnicity_Other | 0.0155 | 1.41458 | 1.06821 | 1.87325 |
| Sex_MALE | 0 | 0.75384 | 0.6845 | 0.8302 |

## ML model performance

To assess performance consistency, we experimented with multiple cross-validation strategies ranging from 2-fold to 10-fold. To prevent data leakage, feature selection was conducted independently within each fold of cross-validation. In every iteration, feature importance was assessed using only the training data. The process included removing multicollinear features via Variance Inflation Factor (VIF ≥ 5), selecting statistically significant predictors using LR (p < 0.05), and identifying the top 10 features based on SHAP values (from XGB) and RF importance scores. A voting mechanism was

then applied to retain features that appeared in at least two of the three methods. This ensured that the test data remained completely unseen during both feature selection and model training.

RF and XGBoost were selected due to their robustness, ability to handle structured survey data, and interpretable outputs. Preliminary evaluations with logistic regression and support vector machines yielded lower accuracy or interpretability, supporting our choice of RF and XGB for final modeling. Regarding model configuration, predefined hyperparameters have been employed for both classifiers. The RF model has been set with n_estimators = 100, and class_weight = 'balanced'. In contrast, the XGB model has been configured with a learning rate of 1, n_estimators=200, max_depth=3, scale_pos_weight=1, and eval_metric='logloss'. These settings have been selected based on prior testing and hyperparameter optimization techniques. Additionaly, to address class imbalance, we used internal built-in mechanisms in the learning algorithms. For the RF model, class_weight='balanced' was applied to adjust the impact of each class during training. In the XGB model, the scale_pos_weight parameter was set to 1 to balance the positive and negative class weights. External oversampling techniques such as SMOTE were tested but ultimately not used, as they did not yield better performance in this dataset and occasionally introduced overfitting.

We applied 10-fold cross-validation as it provided a reliable balance between bias and variance of the model's performance across all data splits. After completing all folds, we computed the average of key performance metrics' accuracy, precision, recall, F1-score, NPV, and ROC AUC in order to summarize the model's generalized performance. Among the evaluated models, XGB yielded superior performance across all metrics. Table 6 provides the 10-fold cross-validation metrics of XGB model. Therefore it was selected as the best performing model for further interpretation and analysis.

Table 7 presents a comparative analysis of performance metrics for two ML models, RF and XGB, used to predict student dropout. The XGB model outperforms the RF model across all evaluated metrics. Specifically, XGB achieves the highest accuracy (0.9441), precision (0.9490), recall (0.9840), F1-score (0.9662), negative predictive value (NPV) (0.9176), and ROC-AUC (0.8774), indicating its superior overall performance in correctly identifying both dropout and non-dropout students. In comparison, the RF model, while still strong, lags slightly behind with an accuracy of 0.9085, precision of 0.9476, recall of 0.9393, F1-score of 0.9434, NPV of 0.7462, and ROC-AUC of 0.8570. These results demonstrate that XGB is more effective and reliable for predicting student dropout.

Fig 5 illustrates the ROC curves for the RF and XGB models used to predict student dropout in Bangladesh. The XGB model achieved a higher AUC of 0.88, indicating superior performance in distinguishing between students who attend school and those who drop out, while the RF model achieved a slightly lower AUC of 0.86, still reflecting strong predictive capability. Both models significantly outperform the random baseline (AUC = 0.5), confirming their reliability in classification tasks. Overall, the XGB model demonstrates better classification power than RF, making it more effective for early identification and intervention strategies aimed at reducing student dropout rates.

**Table 6. XGB 10-Fold Cross-Validation.**

| Fold | accuracy | precision | recall | f1_score | npv | roc_auc |
|------|----------|-----------|--------|----------|-----|---------|
| 1_Fold | 0.943856 | 0.947414 | 0.985792 | 0.966222 | 0.924071 | 0.87271 |
| 2_Fold | 0.939916 | 0.94451 | 0.983191 | 0.963462 | 0.916031 | 0.871823 |
| 3_Fold | 0.951982 | 0.956268 | 0.986466 | 0.971132 | 0.928685 | 0.891331 |
| 4_Fold | 0.944595 | 0.951184 | 0.982488 | 0.966583 | 0.909375 | 0.879762 |
| 5_Fold | 0.946798 | 0.950585 | 0.985749 | 0.967848 | 0.926562 | 0.881982 |
| 6_Fold | 0.942611 | 0.94775 | 0.984027 | 0.965548 | 0.913821 | 0.870719 |
| 7_Fold | 0.945567 | 0.950983 | 0.983607 | 0.96702 | 0.917385 | 0.882795 |
| 8_Fold | 0.942857 | 0.945914 | 0.985303 | 0.965207 | 0.927052 | 0.876783 |
| 9_Fold | 0.939901 | 0.947972 | 0.979648 | 0.963549 | 0.898176 | 0.874589 |
| 10_Fold | 0.94335 | 0.948361 | 0.984342 | 0.966017 | 0.915171 | 0.871738 |

**Table 7. Evaluation of ML models for student dropout prediction.**

| Models | accuracy | precision | recall | f1_score | npv | roc_auc |
|--------|----------|-----------|--------|----------|-----|---------|
| RF | 0.9085 | 0.9476 | 0.9393 | 0.9434 | 0.7462 | 0.8570 |
| XGB | 0.9441 | 0.9490 | 0.9840 | 0.9662 | 0.9176 | 0.8774 |

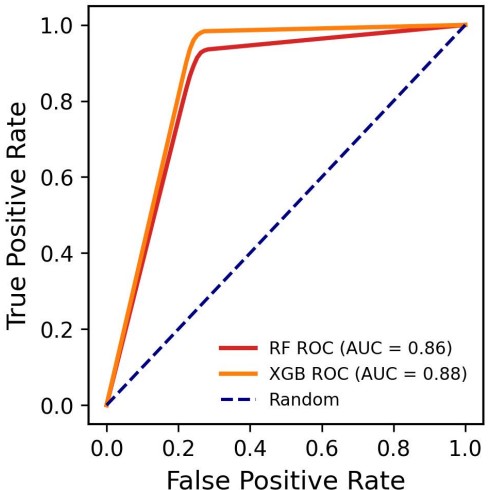

**Fig 5. AUC-ROC curve for different ML models.**

The confusion matrices in Fig 6 compare the performance of the RF and XGB models in predicting school attendance in Bangladesh. Here, the labels represent "NO = 0" for students not attending school and "YES = 1" for those currently attending.

The XGB model demonstrates superior overall classification capability, correctly predicting 3,248 students as attending school (True Positives) and 585 as dropouts (True Negatives). It misclassified only 52 students who were attending as dropouts (False Negatives) and 174 students who were dropouts as attendees (False Positives). These results indicate a strong ability of XGB to detect school-attending students while maintaining relatively low misclassification rates.

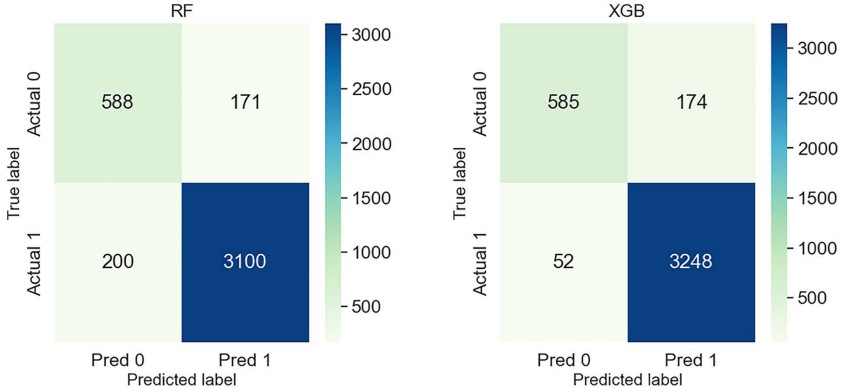

**Fig 6. A bar chart of performance metrics comparing ML models to predict school dropout students in Bangladesh.**

On the other hand, the RF model correctly classified 3,100 students as attendees (True Positives) and 588 as dropouts (True Negatives), which is slightly lower than XGB. However, it also shows higher False Negatives (200) and False Positives (171) compared to XGB. This indicates that RF struggles more with accurately identifying both dropouts and attendees, leading to decreased performance in comparison.

While both models perform reasonably well, the XGB model stands out with fewer misclassifications (total FN + FP = 52 + 174 = 226) compared to RF (200 + 171 = 371). Therefore, based on the confusion matrix evaluation, XGB emerges as the more accurate and balanced model for predicting school attendance, offering a better trade-off between precision and recall in this educational dropout prediction task.

**XAI Interpretation**

We use XGB for SHAP and LIME because it's highly accurate, optimized for fast and exact SHAP computations (Tree SHAP), and handles tabular data efficiently. It ensures meaningful, consistent explanations and supports both global (SHAP) and local (LIME) interpretability with high computational efficiency.

The Fig 7 explains the individual prediction for a student regarding school attendance by quantifying the contribution of each feature. The base value of the model prediction is 1.755, while the final output for this student is 5.501, indicating a strong likelihood of current school attendance. The most significant positive contributor is the student's age (9 years), adding +1.79 to the prediction, as this falls within the typical school-going age. Although the student has not completed any formal grade (Completed_grade_YES = 0), this contributes +0.74, due to positive interactions with other factors. The mother's education level (higher secondary or above) adds +0.62, while the last education grade being Grade 3 adds +0.37, both of which reinforce continued school participation. The father's education (also higher secondary) contributes +0.24. While the student is male (Sex_MALE = 1), this exerts a negative influence (–0.26), indicating that male students, slightly less likely to attend school. The student does not belong to the Chattogram or Dhaka divisions (Division_Chattogram = 0, Division_Dhaka = 0), yet these features contribute positively (+0.19 and +0.12 respectively), due to better attendance rates in other divisions. Finally, the student is from the middle wealth index group (Wealth_index = 2), contributing slightly negatively (–0.06). Overall, the cumulative SHAP values from these features explain why the model strongly predicts that this student is likely to be attending school.

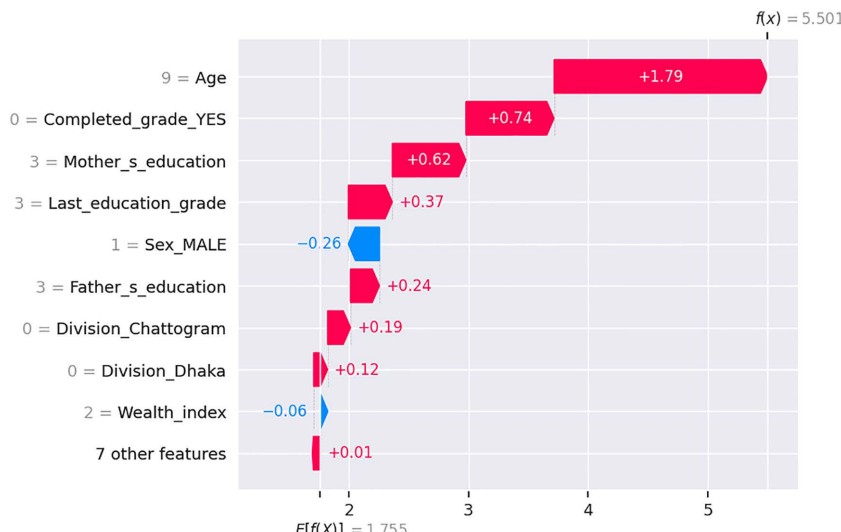

**Fig 7. SHAP waterfall plot showing feature influence on student's school continuation prediction.**

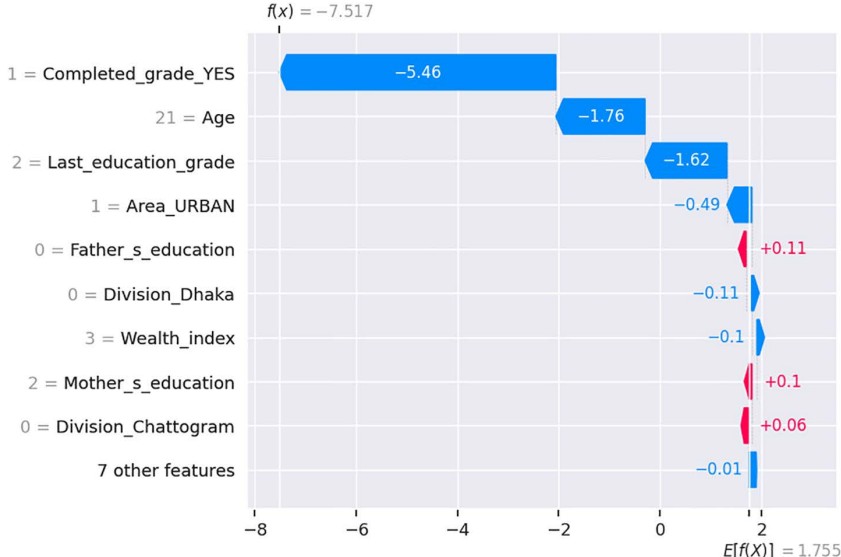 

Fig 8 presents a SHAP-based explanation of the model's base value (expected prediction) is 1.755, while the final output for this student is −7.517, strongly indicating that the student is predicted not to be attending school.

The most influential feature is Completed_grade_YES = 1, contributing −5.46 to the prediction. But the student's age is 21, contributing −1.76. This aligns with the idea that older students with low educational achievement (only grade 2), indicated by Last_education_grade = 2, contributing −1.62 are at greater risk of dropping out. In this case, it reflects a situation where the student is much older but has only completed a few grades suggesting disengagement or delayed progression, which associates with higher dropout likelihood. The Area_URBAN = 1 (urban residence) also negatively impacts the prediction (−0.49), reflecting challenges older students may face in urban environments, such as economic pressures or lack of tailored educational programs. On the positive side, both Father_s_education = 3 (higher secondary) and Mother_s_education = 2 (secondary) contribute +0.11 and +0.10, respectively. This suggests that students with better-educated parents have slightly improved chances of continuing education. However, these contributions are modest and insufficient to offset the strong negative influences. Additionally, Division_Dhaka = 0 and Division_Chattogram = 0 indicate that the student is not from these divisions, meaning they are from one of the other six divisions. These features contribute minor negative impact of not being from depending on other factors. Additionally, the student belongs to the Wealth_index = 3 group (the fourth wealth quintile), which contributes −0.10. This indicates that in this context, being from an upper-middle wealth bracket does not necessarily guarantee continued attendance, due to other overriding socioeconomic or contextual factors. In summary, the SHAP plot shows that a combination of older age, limited educational attainment (grade 2), and urban residence, along with the absence of strong socioeconomic or regional advantages, drives the model's prediction toward non-attendance.

Fig 9 illustrates that the model predicts a high probability (99%) for this 17-year-old female student to continue her education, with only a 1% chance of dropping out. The most infuencing factor behind this prediction is that she has not successfully completed her current grade (Completed_grade_YES = 0), which contributes substantially (0.73) toward continued enrollment. She is currently studying in Grade 11 (Last_education_grade = 11), indicating that she is approaching higher secondary completion, an educational milestone that typically signals strong academic persistence. Her age being over 16 also supports the likelihood of staying in school, particularly when paired with academic consistency. The

**Fig 8. SHAP waterfall plot showing feature influence on student's school dropping out prediction.**

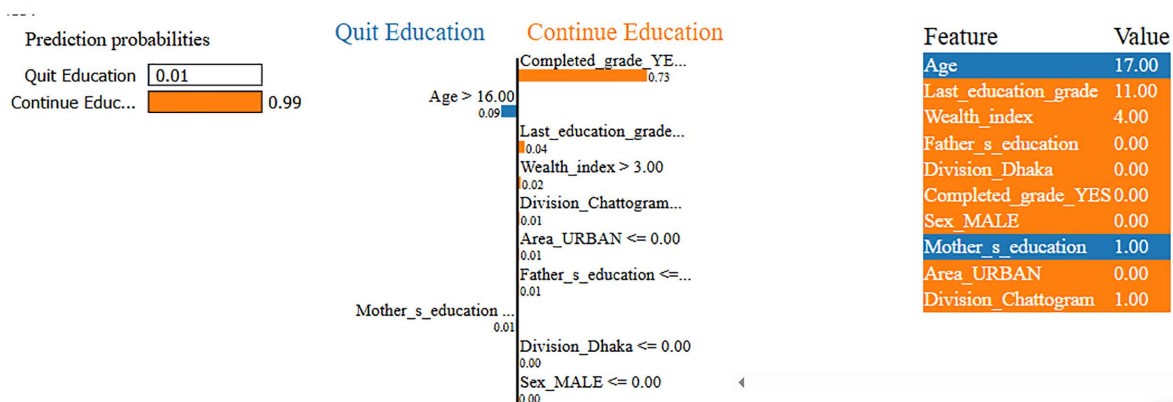

**Fig 9. LIME Analysis of Variable Impact on the Prediction of Continuing Education.**

household's wealth index is categorized as "Richest" (Wealth_index = 4), which adds moderate support to her continued participation. Additional factors, such as residing in the Chattogram division and in a rural area, as well as having a mother with primary education, contribute slightly toward the positive outcome. While her father has no formal education, the factors show negligible influence in this case. Overall, the model's confident prediction is strongly driven by her completed grade status, current academic level, and favorable socioeconomic conditions, all of which reinforce her continued engagement in education despite minor risk factors.

Fig 10 presents a case where the model predicts a 100% probability that this 17-year-old male student will quit education, with virtually no likelihood of continuation. The strongest contributing factor is that the student has completed (Completed_grade_YES = 1) his most recently enrolled grade Grade 2. Within this specific context, appears to signal an endpoint in his academic journey suggesting that he will not proceed to the next level. This interpretation is reinforced by his age being above 16 while currently only having completed Grade 2, reflecting a substantial age-grade mismatch. Such disparity often indicates delayed academic progress and is commonly associated with increased dropout risks. Further compounding the situation, the student belongs to the lowest wealth category (Wealth_index = 0), suggesting severe economic hardship. His parents also have limited educational backgrounds his father completed secondary education and his mother completed only primary limiting the academic support system at home. In addition, he resides in a rural

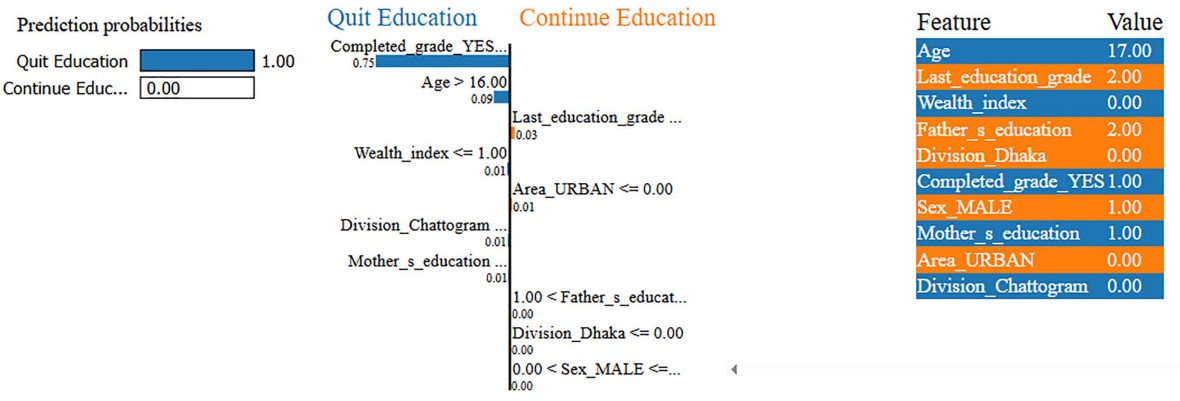

**Fig 10. LIME Interpretation of Variables Influencing Dropout Prediction.**

area outside major administrative divisions such as Dhaka and Chattogram, regions that typically offer more educational infrastructure and opportunities. While grade completion is generally a positive signal, in this case, it appears to represent a stopping point rather than a step forward. Taken together, these factors age-grade mismatch, poverty, limited parental education, and rural location converge to produce a highly confident prediction that the student will exit the education system.

## Policy implication

This hybrid feature selection framework provides a blueprint for future educational research and policy analytics, especially in contexts where interpretability and generalizability are critical. The findings can directly guide targeted dropout interventions, especially for vulnerable groups identified through the model. Fig 11 illustrates a comprehensive, data-driven framework aimed at predicting and preventing school dropouts among children. The process begins at the grassroots level, where families with school-aged children interact with NGOs or educational institutions that collect relevant data on students' educational progress, demographic background, parental influence, and socioeconomic conditions. This data is systematically entered and processed to prepare it for ML analysis. Using predictive algorithms such as RF and XGB, the model identifies students at high risk of dropping out. To ensure transparency in decision-making, XAI

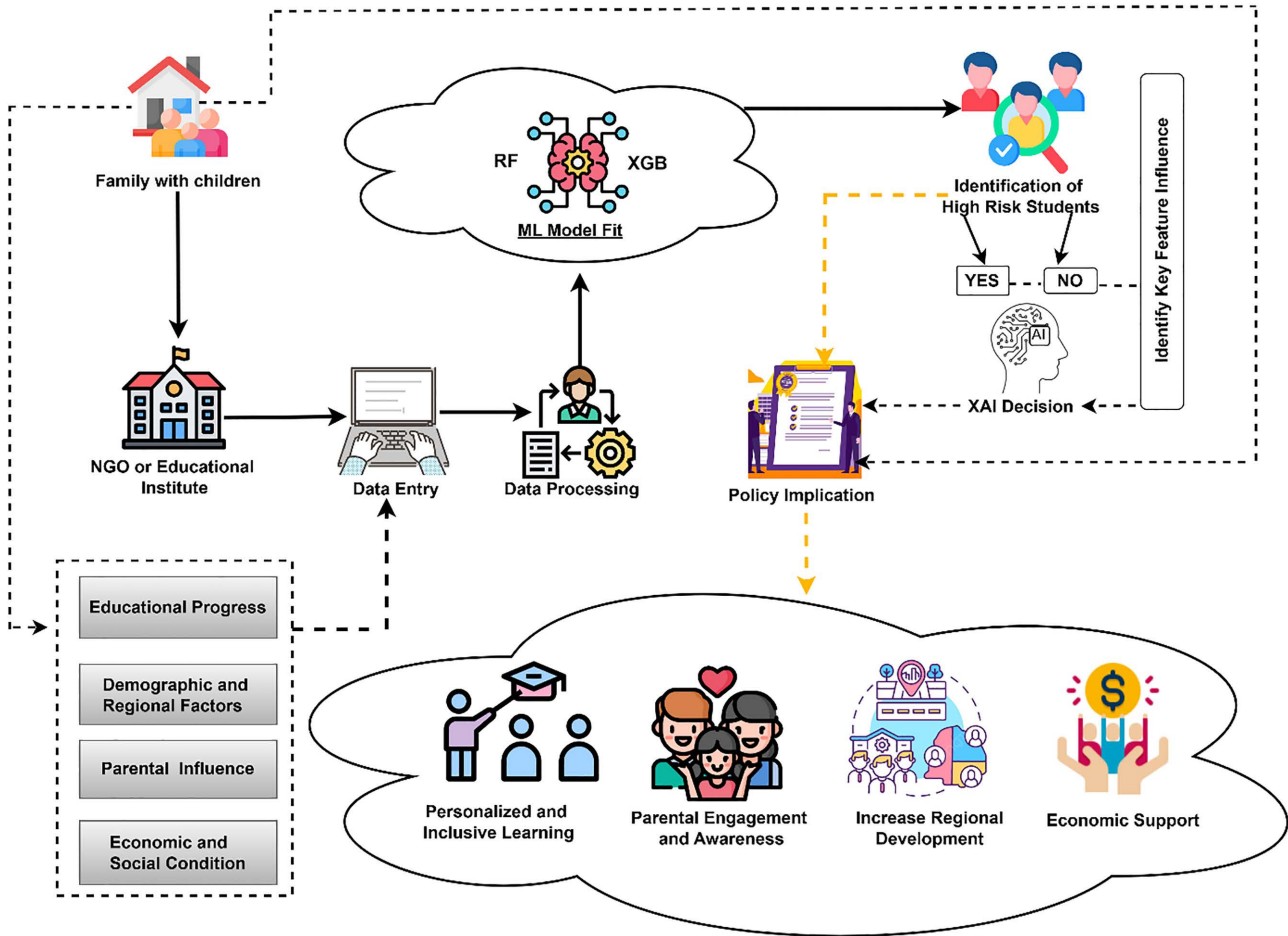

**Fig 11. Data-Driven Framework for School Dropout Prediction and Prevention Policies.**

techniques are applied to highlight the most influential factors driving the dropout risk. Students flagged as high risk are then targeted through policy implications informed by these insights. Suggested interventions include personalized and inclusive learning strategies, parental engagement programs, regional development initiatives, and economic support mechanisms. The framework emphasizes a cycle of informed action starting from data collection to AI-driven prediction, followed by targeted, evidence-based interventions, ensuring that education systems can proactively support vulnerable students and significantly reduce dropout rates.

Based on our proposed model, the following steps can guide policy decisions aimed at lowering student dropout rates:

i. Personalized and Inclusive Learning Strategies: These approaches adjust teaching to match each student's needs, skills, and interests. Since every student learns in their own way, personalized learning gives extra help to those who are struggling. Inclusive strategies make sure that students with older ages also get the support they need. Such an approach keeps students more interested in education and builds their opportunities, which makes them less likely to drop out.

ii. Parental Engagement Programs: Involving parents in their children's education creates a strong support system at home. Programs such as awareness campaigns, parent-teacher meetings, and counseling sessions help families understand the value of education, recognize signs of academic struggle, and encourage their children to stay in school. When lower-educated parents are actively involved, students tend to show better attendance, higher motivation, and continued education.

iii. Regional Development Initiatives: Disparities in educational infrastructure, resources, and teacher quality across different regions (divisions) often contribute to higher dropout rates. By improving school facilities, increasing access to qualified teachers, and ensuring the availability of learning materials in these areas, regional development initiatives make education more accessible and attractive, reducing the need for students to leave school prematurely.

iv. Economic Support: Financial hardship is one of the common reasons for school dropouts. Providing economic support in the form of scholarships, free school meals, uniforms, or transportation reduces the burden on families and encourages continued enrollment. For many households, particularly those in low-income settings, this assistance can make the difference between sending a child to school or finding employment.

While the nationally broad and domain-rich dataset enhances generalizability, it also introduces complexities such as noise, potential biases, and computational overhead. Adding features that seem important in theory but aren't significant could lead to unnecessary repetition in the model or interactions that this study didn't fully address. Furthermore, this study may overlook certain influencing factors that require further analysis in future work.

## Comparison with other works

Table 8 provides a comparative analysis of different studies on student dropout prediction, clearly demonstrating how the current research stands out in terms of scope, methodology, and technological depth. Unlike previous works that were limited to specific education levels (e.g., university or secondary education), our study uniquely targets a comprehensive age group of 6–24 years, capturing the entire education timeline from early childhood to young adulthood. While earlier studies [1,3,31], and [32] employed conventional ML models such as DT, Logistic Regression, and Bayesian Belief Networks, they often suffered from limited sample sizes, a lack of statistical validation, and minimal use of XAI. For instance, only one prior study [31] applied XAI, and none combined it with statistical analysis for better understanding. In contrast, our work integrates both advanced ML models (RF, XGB) and XAI methods (SHAP, LIME), ensuring not only high prediction accuracy (94.4%) but also clear interpretation of the contributing factors. The result is further backed by a comprehensive statistical analysis, which strengthens the credibility of findings. Moreover, your model was trained on a large-scale dataset of 71,531 samples from the 2019 Bangladesh MICS survey, making it more generalizable and reliable than studies

**Table 8. Comparison of student dropout prediction models' performance and methodologies with previous works.**

| Ref. | Objective | Methodology | Dataset | ML Performance | Statistical Analysis | XAI Uses |
|---|---|---|---|---|---|---|
| [1] | Predict dropout in Systems Engineering in a Colombian university | DT, LR, NB, Watson Analytics | 802 students data | DT: 94% (AUC) | [×] | [×] |
| [3] | Predict freshman dropout using Bayesian Belief Networks | Bayesian Belief Network (BBN), SMOTE | 36,461 records for public university students | BBN: 84% (AUC) | [×] | [×] |
| [31] | Predict SSC/HSC performance using explainable AI | ETC, RF, LR, NB, DT, and SVM, SHAP framework | Not mentioned | RF: Accuracy 82.23% for SSC, 86.89% for HSC result | [×] | [✓] |
| [32] | Predict dropout in higher education using ML | DT, NB, KNN, GB Trees, DL | Models built on data from 15,825 students | GB Trees: 0.808 (AUC), DL 0.811 (AUC) | [×] | [×] |
| This work | Predict student dropout from 3–24 years. | RF, XGB, SHAP, LIME | Models run on 71531 samples from the MICS 2019 Bangladesh dataset. | XGB: Accuracy 94.4%, precision 0.9490,recall 0.9840, f1_score 0.9662,npv 0.9176, roc_auc 0.8774 | [✓] | [✓] |

using smaller or unspecified datasets. The integration of XAI sets our work apart, allowing policymakers and educators to understand why certain students are at risk of dropping out, rather than just knowing who is at risk.

## Limitations and future work

While this study offers valuable insights into the underlying causes of school dropout, it has some limitations that must be acknowledged. First, it used data from a single cross-sectional survey (MICS 2019 Bangladesh). The findings may not generalize to other contexts or future populations. Second, while advanced feature selection techniques were applied, some potentially important variables may not have been captured in the dataset. Third, survey weights were not applied during predictive modeling, as the focus was on model performance rather than generating population-representative estimates. The models used in this study (Random Forest and XGBoost) were chosen for their interpretability and performance on tabular data. However, deep learning methods, which may reveal more complex patterns, were not explored. Future research could investigate the use of deep learning approaches and integrate longitudinal data to better capture causal relationships, and validate models on other datasets in order to improve generalizability. Collaboration with policymakers is also recommended to ensure the models can induce practical interventions.

## Conclusion

This research successfully identifies key factors influencing student dropout in Bangladesh, leveraging the MICS 2019 dataset and statistical analysis. Furthermore, we have used advanced ML models and techniques to forecast student dropout. The analysis highlights that prior school attendance, household education level, and age of students are the most influential predictors of school retention. Among the ML models applied, the XGB model emerged as the most effective classifier, combining high accuracy and efficiency, making it an ideal choice for real-time student dropout prediction. Additionally, the integration of XAI techniques like SHAP and LIME ensured that the model's predictions were transparent, thus building trust in the results. These findings provide valuable insights for educators and policymakers, offering a data-driven approach to develop targeted strategies to reduce dropout rates and improve student retention. While the study offers a strong foundation for understanding dropout patterns, it also acknowledges the need for further research to explore other possible factors and to apply the models in different regions for broader applicability. By addressing these issues, the findings of this research have the potential to contribute significantly to shaping educational policy and fostering improved socio-economic outcomes in Bangladesh.

## Acknowledgments

The authors would like to express their sincere gratitude to the Bangladesh Bureau of Statistics (BBS) and UNICEF for providing access to the 2019 Multiple Indicator Cluster Survey (MICS) dataset, which served as the foundation for this study. The availability of high-quality, nationally representative data made it possible to conduct a comprehensive and impactful analysis of the factors influencing student dropout in Bangladesh.

## Author contributions

**Conceptualization:** Mst. Rokeya Khatun.

**Data curation:** Mst. Rokeya Khatun.

**Formal analysis:** Mst. Rokeya Khatun, Mithila Akter Mim.

**Investigation:** Mst. Rokeya Khatun.

**Methodology:** Mst. Rokeya Khatun.

**Software:** Mithila Akter Mim.

**Supervision:** Mst. Rokeya Khatun, Md. Minoar Hossain.

**Validation:** Mst. Rokeya Khatun, Mithila Akter Mim, Md. Mahadi Tasin.

**Visualization:** Mst. Rokeya Khatun, Mithila Akter Mim.

**Writing – original draft:** Mst. Rokeya Khatun, Mithila Akter Mim, Md. Mahadi Tasin.

**Writing – review & editing:** Mst. Rokeya Khatun, Mithila Akter Mim, Md. Mahadi Tasin, Md. Minoar Hossain.

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
