## [Decision Letter · Decision Letter 0]

26 May 2025

Dear Dr. Khatun,

Thank you for submitting your manuscript to PLOS ONE. After careful consideration, we feel that it has merit but does not fully meet PLOS ONE’s publication criteria as it currently stands. Therefore, we invite you to submit a revised version of the manuscript that addresses the points raised during the review process.

We look forward to receiving your revised manuscript.

Kind regards,

Elochukwu Ukwandu, PhD

Academic Editor

PLOS ONE

Journal Requirements:

Reviewers' comments:

Reviewer's Responses to Questions

**Comments to the Author**

1. Is the manuscript technically sound, and do the data support the conclusions?

Reviewer #1: Partly

Reviewer #2: Yes

2. Has the statistical analysis been performed appropriately and rigorously?

Reviewer #1: Yes

Reviewer #2: Yes

3. Have the authors made all data underlying the findings in their manuscript fully available?

Reviewer #1: Yes

Reviewer #2: Yes

4. Is the manuscript presented in an intelligible fashion and written in standard English?

Reviewer #1: Yes

Reviewer #2: Yes

Reviewer #1: 1. Clearly define the sample inclusion criteria—especially how school-age was defined (e.g., why include 3–5 year-olds?).

2. Clarify whether model validation used a hold-out set or K-fold cross-validation (and specify the value of K).

3. Ensure feature selection (logistic p-values, RF/SHAP importance) was done inside cross-validation folds to avoid data leakage.

4. Justify why Ethnicity was dropped despite statistical significance, while Mother’s education was retained despite insignificance.

5. Describe how SHAP/RF “important” features were selected (top N, threshold, etc.).

6. Quantify and explain how missing values and outliers were handled (how many rows removed or imputed?).

7. Justify use of label encoding for categorical variables like Division; consider one-hot encoding for nominal categories.

8. Provide details on hyperparameter tuning for XGB and RF (number of trees, learning rate, etc.), or clarify if default settings were used.

9. State whether class imbalance mitigation (e.g., SMOTE or class weights) was applied.

10. Report and discuss the actual dropout rate (i.e., % of students not attending school in the dataset).

11. Clarify the dependent variable's coding (dropout = 1?) to align interpretations of odds ratios.

12. Add a correlation matrix or VIF scores to support low multicollinearity claims among predictors.

13. Indicate whether survey weights from MICS were used; if not, briefly justify.

14. Reduce textbook-style definitions of OR/p-values and instead interpret findings (e.g., “OR = 1.05 means…”).

Reviewer #2: 1. In the abstract, it is recommended to explain how statistical analysis is integrated with machine learning models (XGBoost and Random Forest) and explainable artificial intelligence (XAI) tools such as SHAP and LIME.

2. The literature reviews are not sufficiently thorough, and the gaps to be addressed are not clearly defined. Add studies using deep learning.

3. It is suggested to explain how the variables were selected and how imbalances in the data were addressed (such as using SMOTE techniques). It is important for the authors to explain why the models (XGBoost and Random Forest) were chosen over others.

4. Add a clear explanation for each graph and how it helps understand the result.

5. It is recommended to add a clear section titled "Limitations of the Study and Future Work" that addresses the methodological or technical limitations of the current study and suggests future paths for expansion, including the application of deep learning techniques.

**Do you want your identity to be public for this peer review?** For information about this choice, including consent withdrawal, please see our Privacy Policy

Reviewer #1: No

Reviewer #2: No

---

## [Author Response · Author response to Decision Letter 1]

9 Aug 2025

Responses to reviewers:

The authors would like to thank the reviewers for their precious time and valuable suggestions. We have carefully addressed and updated all the changes in the original manuscript. The following section provides point-by-point responses corresponding to all review comments.

Reviewer #1:

1. Clearly define the sample inclusion criteria—especially how school-age was defined (e.g., why include 3–5 year-olds?)

Response: Thank you for this important observation. We have revised the “Data Preprocessing” subsection to clarify the inclusion criteria.

To ensure data quality, we excluded samples with missing target values and responses marked as "Don't know" or "No response." The sample population was defined using age groups from the UNICEF MICS 2019, which categorizes school-going children aged 6–24, aligning with global education standards. Children aged 3–5 were initially included as "school-age" based on the Early Childhood Education (ECE) module of MICS, recognizing this stage as critical for early learning and pre-primary access. Education typically progresses from primary (ages 6–10), to lower secondary (11–13), upper secondary (14–15), higher secondary (16–17), and tertiary or vocational training (18–24), though variations may occur due to delayed entry, repetition, or dropout. As ECE or early learning is not formal education and as one cannot drop out of informal education like ECE, school age was defined to be 6-24, excluding the age range of ECE (3-5).

2. Clarify whether model validation used a hold-out set or K-fold cross-validation (and specify the value of K).

Response: We clarify that model validation was performed using K-fold cross-validation with K = 10. Detailed information has been included in the “ML Model Performance” subsection of the manuscript.

3. Ensure feature selection (logistic p-values, RF/SHAP importance) was done inside cross-validation folds to avoid data leakage.

Response: We thank the reviewer for pointing this out. We confirm that feature selection (logistic regression p-values, Random Forest importance, and SHAP values from XGBoost) was performed strictly within each cross-validation fold to avoid data leakage. This process ensured that feature selection was based only on the training data in each fold, maintaining the integrity of model validation. We have clarified this procedure in the revised “ML model performance” section.

4. Justify why Ethnicity was dropped despite statistical significance, while Mother’s education was retained despite insignificance.

Response: Thank you for the comment. Ethnicity was dropped despite statistical significance, it’s practical contribution to model performance was minimal, which affected model stability. In contrast, Mother’s education was retained because, despite insignificance, it is a key factor supported by prior research and important for interpretation. This rationale is now clarified in the manuscript “Feature Selection” section.

Although Ethnicity was statistically significant (p = 0.012) in the logistic regression analysis, it was dropped because it showed very low predictive importance in both the RF and SHAP analyses ranking. This indicates that, despite statistical significance, its practical contribution to model performance was minimal. In contrast, Mother’s education, though statistically insignificant (p = 0.536), was retained due to its comparatively higher importance in both ML models and its strong theoretical and contextual relevance. Prior research consistently highlights maternal education as a key determinant of school dropout. Including it helps preserve model interpretability and aligns with domain knowledge, ensuring the model remains meaningful for education-related policy and intervention design.

5. Describe how SHAP/RF “important” features were selected (top N, threshold, etc.).

Response: The value of N=10 and threshold=5 have been added in the “ML Model Performance” section.

The process included removing multicollinear features via Variance Inflation Factor (VIF ≥ 5), selecting statistically significant predictors using logistic regression (p < 0.05), and identifying the top 10 features based on SHAP values (from XGBoost) and Random Forest importance scores. A voting mechanism was then applied to retain features that appeared in at least two of the three methods. This ensured that the test data remained completely unseen during both feature selection and model training.

6. Quantify and explain how missing values and outliers were handled (how many rows removed or imputed?).

Response: We have quantified and explained the handling of missing values and outliers, including the number of rows removed and imputed in the “Data Preprocessing” section of the manuscript.

Initially, the dataset comprised of 48,412 samples; after filtering out these entries, 7,808 rows were discarded, resulting in a final set of 40,604 samples. For categorical features, we replaced missing values with the mode, while for numerical features, we used the mean. After that, we reviewed each feature for mixed data types and converted them into their proper formats casting values to floats, integers, or strings as needed. This step ensured that every column followed a consistent type convention throughout the dataset. Following this, we removed outliers, further refining the dataset to 11 features with 38,213 samples.

7. Justify use of label encoding for categorical variables like Division; consider one-hot encoding for nominal categories.

Response: Thank you for the insightful comment. Among the features we have applied one-hot encoding to the nominal variables: ‘Area’, ‘Division’, ‘Completed_grade’, ‘Ethnicity’, and ‘Sex’. The remaining categorical variables, such as ‘Wealth_index’, ‘Mother_s_education’, and ‘Father_s_education’, were label encoded due to their ordinal nature. We have clarified this point in the revised manuscript in the “Data Preprocessing” section.

8. Provide details on hyperparameter tuning for XGB and RF (number of trees, learning rate, etc.), or clarify if default settings were used.

Response: We acknowledge the comment and confirm that details regarding hyperparameter settings for both XGB and RF models have been added in the “ML Model Performance” section.

9. State whether class imbalance mitigation (e.g., SMOTE or class weights) was applied.

Response: We appreciate the reviewer’s observation. Yes, class imbalance was addressed using internal weighting mechanisms. Specifically, class_weight='balanced' was applied in the Random Forest model, and scale_pos_weight=1 was set in the XGBoost model. These approaches were preferred over external techniques like SMOTE, as preliminary experiments with SMOTE did not improve performance and, in some cases, led to overfitting. This has now been clarified in the revised “ML model performance” section.

10. Report and discuss the actual dropout rate (i.e., % of students not attending school in the dataset).

Response: We have reported and discussed the actual dropout rate, the percentage of students not attending school in the manuscript in the “Dataset” section. According to the dataset distribution, 81.28% of students attended school, whereas did not, indicating a dropout rate of approximately 18.72%.

11. Clarify the dependent variable's coding (dropout = 1?) to align interpretations of odds ratios.

Response: Thank you for your insightful comment. We have clarified the coding of the dependent variable, Attend_school_thisYear, in the “Statistical Measurements” section. Specifically, the variable was coded as 1 for children not attending school (dropout) and 0 for those currently attending school. This coding was used consistently throughout all statistical analyses and machine learning models. We also elaborated on the interpretation of odds ratios based on this coding, where an odds ratio greater than 1 indicates an increased likelihood of dropout, and an odds ratio less than 1 indicates a decreased likelihood. This clarification ensures consistent and accurate interpretation of model outputs across the study.

12. Add a correlation matrix or VIF scores to support low multicollinearity claims among predictors.

Response: As advised, we have added the VIF scores to demonstrate low multicollinearity among predictors. This has been incorporated in the “Feature Selection” section and the details are presented in Table 4.

13. Indicate whether survey weights from MICS were used; if not, briefly justify.

Response: We have addressed this point in the “Dataset” section by briefly indicating that MICS survey weights were not used and providing a justification for this decision.

As the aim of this study was to develop predictive models and identify key factors associated with school dropout rather than to derive nationally representative estimates, the survey weights provided by MICS were not applied. While such weights are crucial for calculating population-level statistics, they are not always necessary or appropriate in predictive modeling tasks.

14. Reduce textbook-style definitions of OR/p-values and instead interpret findings (e.g., “OR = 1.05 means…”).

Response: As recommended, we have reduced textbook-style definitions and provided practical interpretations of key statistical metrics. Specifically, in the “Statistical Measurements” section.

Reviewer #2:

1. In the abstract, it is recommended to explain how statistical analysis is integrated with machine learning models (XGBoost and Random Forest) and explainable artificial intelligence (XAI) tools such as SHAP and LIME.

Response: We appreciate the reviewer for this valuable suggestion. The explanation on the integration of statistical analysis, machine learning models and explainable artificial intelligence tools in the abstract has been enhanced and updated in an attempt to make the integration even more interpretable.

2. The literature reviews are not sufficiently thorough, and the gaps to be addressed are not clearly defined. Add studies using deep learning.

Response: We acknowledge the reviewer’s concerns. The research gap in the literature review has been addressed through the addition of studies implementing deep learning techniques and the overall enhancement of the literature review in “Introduction” section.

3. It is suggested to explain how the variables were selected and how imbalances in the data were addressed (such as using SMOTE techniques). It is important for the authors to explain why the models (XGBoost and Random Forest) were chosen over others.

Response: We appreciate the reviewer’s suggestion. The “Feature Selection” section has been updated in order to clearly explain the variable selection strategy, which combined statistical and model-based criteria within each cross-validation fold. We also clarified how class imbalance was addressed using internal model techniques, and provided a brief justification for the selection of Random Forest and XGBoost, in section “Machine Learning Models”.

4. Add a clear explanation for each graph and how it helps understand the result.

Response: we have carefully revised the manuscript to include clear and detailed explanations for each graph

5. It is recommended to add a clear section titled "Limitations of the Study and Future Work" that addresses the methodological or technical limitations of the current study and suggests future paths for expansion, including the application of deep learning techniques.

Response: We thank the reviewer for this valuable suggestion. In response, we have added a new section titled "Limitations and Future Work" at the end of the Discussion section. This new section acknowledges several methodological limitations of the present work, including the use of non-deep-learning models, reliance on a single dataset (MICS 2019), and the absence of survey weights in predictive modeling. We also discuss future scopes for research, such as exploring deep learning methods, incorporating additional data sources, and validating the models across multiple survey waves to improve generalizability and robustness.

---

## [Decision Letter · Decision Letter 1]

24 Aug 2025

A Hybrid Framework of Statistical, Machine Learning, and Explainable AI Methods for School Dropout Prediction

PONE-D-25-21220R1

Dear Dr. Khatun,

We’re pleased to inform you that your manuscript has been judged scientifically suitable for publication and will be formally accepted for publication once it meets all outstanding technical requirements.

Kind regards,

Elochukwu Ukwandu, PhD

Academic Editor

PLOS ONE

Additional Editor Comments (optional):

Reviewers' comments:

Reviewer's Responses to Questions

**Comments to the Author**

Reviewer #2: All comments have been addressed

Reviewer #3: All comments have been addressed

2. Is the manuscript technically sound, and do the data support the conclusions?

Reviewer #2: Yes

Reviewer #3: (No Response)

3. Has the statistical analysis been performed appropriately and rigorously?

Reviewer #2: Yes

Reviewer #3: (No Response)

4. Have the authors made all data underlying the findings in their manuscript fully available?

Reviewer #2: Yes

Reviewer #3: (No Response)

5. Is the manuscript presented in an intelligible fashion and written in standard English?

Reviewer #2: Yes

Reviewer #3: (No Response)

Reviewer #2: (No Response)

Reviewer #3: Accept

Accept

AcceptAcceptAcceptAcceptAcceptAcceptAcceptAcceptAcceptAcceptAcceptAcceptAcceptAcceptAcceptAcceptAcceptAcceptAcceptAcceptAcceptAcceptAcceptAcceptAcceptAcceptAcceptAcceptAcceptAcceptAcceptAcceptAcceptAcceptAcceptAcceptAcceptAcceptAcceptAcceptAcceptAcceptAcceptAcceptAcceptAcceptAcceptAcceptAcceptAcceptAcceptAcceptAcceptAcceptAcceptAcceptAcceptAcceptAcceptAcceptAcceptAcceptAcceptAcceptAcceptAcceptAcceptAcceptAcceptAcceptAcceptAcceptAcceptAccept

**Do you want your identity to be public for this peer review?** For information about this choice, including consent withdrawal, please see our Privacy Policy

Reviewer #2: No

Reviewer #3: No

---

## [Editor Report · Acceptance letter]

PONE-D-25-21220R1

PLOS ONE

Dear Dr. Khatun,

I'm pleased to inform you that your manuscript has been deemed suitable for publication in PLOS ONE. Congratulations! Your manuscript is now being handed over to our production team.

Kind regards,

on behalf of

Dr. Elochukwu Ukwandu

Academic Editor

PLOS ONE